# Interaction between Cognitive Reserve and Biomarkers in Alzheimer Disease

**DOI:** 10.3390/ijms21176279

**Published:** 2020-08-30

**Authors:** Elena Carapelle, Ciro Mundi, Tommaso Cassano, Carlo Avolio

**Affiliations:** 1Department of Neuroscience, United Hospital of Foggia, 71100 Foggia, Italy; cmundi@ospedaliriunitifoggia.it (C.M.); carlo.avolio@unifg.it (C.A.); 2Department of Clinic and Experimental Medicine, University of Foggia, 71100 Foggia, Italy; tommaso.cassano@unifg.it; 3Department of Medical and Surgical Sciences, University of Foggia, 71100 Foggia, Italy

**Keywords:** cognitive reserve, Alzheimer disease, CSF biomarkers

## Abstract

Patients with comparable degree of neuropathology could show different cognitive impairments. This could be explained with the concept of cognitive reserve (CR), which includes a passive and an active component. In particular, CR is used to explain the gap between tissue damage and clinical symptoms that has been observed in dementia and, in particular, in patients affected by Alzheimer disease (AD). Different studies confirm brain neuroplasticity. Our preliminary study demonstrated that AD patients with high education showed a CR inversely associated with glucose uptake measured in fluorodeoxyglucose positron emission tomography (FDG-PET), whereas the inverse correlation was observed in AD patients with low education. In other words, our findings suggest that CR compensates the neurodegeneration and allows the maintenance of patients’ cognitive performance. Best understanding of the concept of CR could lead to interventions to slow cognitive aging or reduce the risk of dementia.

## 1. Introduction

Alzheimer disease (AD) is a neurodegenerative disorder due to pathological accumulation of β-amyloid. Clinically it is characterized by impaired short-term memory and a progressive decrease in cognitive abilities [1]. From the anatomopathological point of view, it is associated with a continuing accumulation of senile plaques and neurofibrillary tangles, which are responsible for neuronal death. Brain damage concerns, at first, medial temporal lobes [2] and subsequently involves lateral temporal, parietal, and frontal regions [3]. Several studies have investigated the sensitivity of specific biomarkers for AD in order to discriminate the different stages of the disease and provide prognostic measurements [4,5]. Cerebral glucose uptake (GU), measured through fluorodeoxyglucose positron emission tomography (FDG-PET), and cerebrospinal fluid (CSF) biomarkers are considered indicators for AD in vivo. FDG-PET, in particular, detects not only cerebral metabolism, but also synaptic activity and is, therefore, reputed to be an indirect measure of neuronal integrity [6,7]. Based on FDG-PET imaging pattern, AD is characterized by a specific regional reduction of GU at the posterior cingulate level and in the parieto-temporal cortex [8,9]. Symptoms of AD essentially do not occur in the absence of hypometabolism, the amount of which typically reflects the severity of cognitive impairment [7]. Thanks to their diagnostic accuracy, CSF biomarkers are also used in the clinical diagnosis of AD, and include markers for fibrillar neurodegeneration (such as total tau protein (Tau) and tau protein phosphorylated at threonine 181 (181p-tau)) and the deposition of β-amyloid_1-42_ [Aβ_1-42_]. Several studies have validated the use of low levels of CSF Aβ_1-42_ as a marker for AD, although there is variability in the establishment of cutoff values for normality [10,11]. 

## 2. Classification and the Use of New Biomarkers in Alzheimer Disease

Revised NINCDS-ADRDA diagnostic revised criteria have highlighted a common framework regarding AD definition and its staging [12,13]. This is why the National Institute of Aging and Alzheimer’s Association (NIA-AA), 2018, has created a fresh approach scheme. These new recommendations should be considered as research framework and not as diagnostic criteria or guidelines for diagnosis [14]. The need to read them as research framework derives from the common interest in validating and modifying data, before they can be used in usual clinical practice [14]. Definitions can be agreed upon on the basis of imaging and CSF biomarkers, which are widely used in research on AD and brain aging. The scheme (labeled as ATN) recognizes three general groups of biomarkers according to the nature of the pathological process measured [14]. Biomarkers for amyloid plaques (indicated by “A”) are represented by the cortical ligands in amyloid PET [15,16] or by low levels of CSF-Aβ_42_ [17,18]. Biomarkers for tau fibrils (labeled “T”) are portrayed by levels in the CSF of phosphorylated tau (P-tau) and cortical tau ligands at PET [19]. Biomarkers for neurodegeneration or neuronal lesion (labeled “N”) are the total tau at CSF (T-tau) [20], FDG-PET hypometabolism, and MRI atrophy patterns [20]. The suitable mixture of biomarkers’ image may be chosen based on the available resources; for example, when lumbar puncture and MRI are accessible but not PET, researchers can opt for CSF-Aβ_42_ and p-tau, respectively, as A and T biomarkers and MRI as N biomarker. 

Each group of ATN biomarkers results in eight different ATN “biomarker profiles”: A + TN-, A + T + N +, etc., based on definitions. The ATN biomarker system assigns a subject to each of the three "biomarker categories": (1)Individuals with normal AD biomarkers.(2)Those in the continuum of AD (divided into “Alzheimer pathological change” and “AD”).(3)Those with a normal amyloid biomarker but with abnormal T, N, or both. This latter biomarker profile implies evidence of one or more neuropathological processes other than AD and has been labeled as “suspected non-Alzheimer pathophysiology”(SNAP) [21].

An individual with biomarker evidence of Aβ deposition alone (abnormal PET or low CSF levels of Aβ_42_ or low ratio 42/40) with a normal tau biomarker would be associated to the “Alzheimer pathological change” label. The term “AD” will be applied if biomarker evidence of both Aβ and pathological tau are present. “Alzheimer pathological change” and “AD” are not considered separate entities but as the earlier and later stages of the “Alzheimer continuum” (an umbrella term that includes both Alzheimer pathological change and AD). 

Neurodegenerative/neuronal biomarkers and cognitive symptoms, none of which are specific to AD, are used only as stage severity to avoid defining the presence of the AD continuum [20].

From literature data [21], the use of amyloid PET can be promising in two types of scenarios: (1) To exclude AD in specific cases where it is difficult to formulate a differential diagnosis, for example, AD versus fronto-temporal dementia (FTD) and (2) in order to create a good number of clinical stages that aim to improve treatments or prevention strategies for AD, allowing patients to be enrolled in accordance not only with clinical and epidemiological criteria, but also with biological markers. It is important to consider in the diagnosis of AD the possibility of performing a liquor dosage of substances with a potential pathogenetic role in dementia, such as Aβ, Tau, and P-tau.

It is established that the pathological process in the brain of AD patients begins more than a decade before the first symptoms [2]. Thanks to the description of the temporal dynamics of biomarker levels in relation to cognitive changes, it has been possible to realize a hypothetical model of the AD course based on the levels of liquor markers in the different stages [22]. The liquor markers used are the total tau protein (tau), which reflects the magnitude of the neuro-axonal degeneration, the P-tau, correlated with neurofibrillary degeneration, and the Aβ isoform made of 42 amino acids (Aβ_1-42_), related to senile plaques. Liquor levels of T-tau are 300% higher in patients with AD than in controls [23]. This marker is not considered specific for AD and elevated levels can be found in the CSF of patients with stroke or head trauma; levels even higher are found in patients with Creutzfeldt-Jakob disease [24]. Other types of dementia may also present elevated T-tau values, since this is not considered a specific marker of neuronal and axonal degeneration. 

The tau protein exists in several isoforms and can be phosphorylated on different residues. The most common ELISA test for T-tau identifies all the isoforms, regardless of their phosphorylation status. The most widespread ELISA tests for P-tau measure phosphorylated tau at residues 181 or 231 and the diagnostic performance between the different methods is similar. A multiparameter method for the simultaneous measurement of Aβ_42_, P-tau, and T-tau has also been developed. Specifically, biomarkers’ values can be organized in a two-dimensional graph, with P-tau levels on the x-axis and the IATI index (INNOTEST Amyloid Tau Index) on the ordinate axis, which allows combining, for its turn, the values of Aβ_1-42_ and P-tau with the formula of Hulsteart [25] (IATI = Aβ_1-42_/(240 + 1.18 tau)). The multicenter study conducted by Hulsteart et al. established cutoff values of 1 for IATI (IATI < 1: Suggestive for AD; IATI > 1: Normal) and 61 ρg/mL for p-tau. Several authors then used these parameters in their studies, finding sensitivity values up to 94% in the diagnosis of AD; the lowest levels of specificity were found, however, in the diagnosis of VaD (48%) [26]. Within the population affected by AD there is a subgroup recognized as SNAP (suspected non-AD pathophysiology) [21], in which subjects have normal amyloid markers, but show signs of neurodegeneration different from AD. The combination of liquor biomarkers and neuroimaging can help increase diagnostic accuracy, compared to the methods used individually (see Table 1).

## 3. The Theory of Cognitive Reserve

Despite better knowledge on the anatomopathological substrate of Alzheimer disease, the relationship between pathology and cognitive ability has always been recognized as complex and nonlinear. 

The concept of cognitive reserve (CR) was introduced to explain the gap between the extent of brain tissue damage and clinical symptoms observed on an individual level [27]. According to this hypothesis, AD patients with higher CR than those with lower CR require a greater accumulation of damage from brain pathology in order to show the same amount of cognitive impairment [27]. 

Cerebral aging is characterized by important inter-individual differences [28]. These differences are found not only in the structural [29], metabolic [30], and chemical [31] changes of the brain, but also in the ability to compensate for the losses associated with brain “damage” due to the normal aging process [32] or the onset of a degenerative pathology [33]. Since the late 1980s, numerous studies have shown that: (1) Pathological changes in AD do not necessarily produce clinical manifestations [34,35,36,37] and (2) acquired brain damage of comparable severity may cause different levels of cognitive impairment characterized by differences in the speed of recovery of the disorder [38]. Recent studies on centenarians have also highlighted how, despite the presence of age-dependent disorders, it is possible to live to a very advanced age while remaining functionally independent [39,40]. The nondirect relationship between the degree of severity of brain damage and its clinical manifestations has led us to propose and adopt the notion of reserve. Classically, the reserve is classified into two models, respectively, the cerebral reserve and the CR. In the cerebral reserve model “threshold” [41], the reserve is conceived as a passive process and defined in terms of the amount of damage that the brain can accumulate before translating into a clinical expression. This model specifically refers to brain damage due to both the aging process and acquired and degenerative pathological processes. In the CR model [42,43], however, it is assumed that the brain adopts active methods to counter or compensate for a pathological process through the use of cognitive processes and functional connection networks between neurons.

The concept of brain reserve was initially introduced in the scientific literature to explain how brain injuries typical of degenerative diseases can be diagnosed before the symptoms appear. The first study in which the term “reserve” was used dates back to Katzman [35]. The authors, examining the brains of 137 elderly post-mortem, found that temporal dissociation between extensive pathological damage and their clinical manifestation was only for certain individuals. Unlike individuals with pathological damage and clinical manifestation, those without clinical manifestation had a larger number of neurons. This result led the authors to propose that the absence of clinical manifestation could be due to (1) incipient AD without a significant loss of neurons and (2) a greater “reserve”, thanks to the size of the brain and the greater presence of neurons. More recently, in line with the results of Katzman and Crystal [35,44], Davis and collaborators with a post-mortem study showed that most of the brains of the 59 elderly people who had been followed longitudinally for eight years, while presenting an abundant amount of senile plaques and neurofibrillary tangles, were not characterized by significant cognitive changes [45]. These results suggest that the decrease in the amount of tissue (neurons or synapses) subject to alteration, when exceeded, leads to the manifestation of a clinical or functional disorder, defined as a fixed critical threshold. This means that for the same brain damage, if the threshold is exceeded, it produces the same outcome in each individual and that the differences between individuals at the level of the clinical manifestation are due only to the overall reserve capacity of the brain. Various studies on the prevalence of AD and its incidence confirm the existence of this critical threshold [41]. Starting from this definition, Satz [46] introduced the concept of cerebral reserve capacity, in which particular importance is given to the role of inter-individual differences. In fact, Satz noted that particularly important brain damage could produce a clinical or functional disorder in a patient with less brain reserve capacity, if this exceeds the critical “threshold” of brain damage necessary to produce a behavioral disorder. On the contrary, a patient with a higher brain reserve capacity can maintain a level of functionality for longer because the damage does not reach this critical level. 

Mortimer, Borenstein, Gosche, and Snowdon highlighted how, when the clinical signs of Alzheimer dementia become visible, brain damage can be relatively extensive in individuals with greater brain reserves than individuals with lower brain reserves [47]. In addition, the inter-individual differences in the appearance of symptoms seem to be related to the different rate of loss of the reserve. The reserve quantity, therefore, assumes an intermediate role between the pathology and the clinical manifestation, influencing the severity of the clinical or functional symptoms once the critical threshold is reached. Indeed, pre-existing damage reduces the amount of residual brain reserve that acts as a “buffer resource” and that preserves behavior for a certain period of time from the effects of changes or from brain damage. On an empirical level, the cerebral reserve is currently operationalized with “direct” variables that reflect the more purely structural aspects of the brain. For example, individuals with a larger head circumference [48], a larger brain volume [48], or a stronger synaptic density [49] have been shown to manifest deficits associated with degenerative processes later than individuals with values lower than the indicators, thanks to a neural substrate sufficient to support nonpathological functioning. Nonetheless, from the moment the pathology is diagnosed, and the buffer reserve runs out, the evolution of behavioral disorders and deficits becomes faster [49,50,51]. Although it is undeniable that individuals differ in the amount of brain reserve available (or in their brain reserve capacity), a limitation of this research was to neglect the qualitative differences between individuals in the use of available resources. In fact, the cerebral reserve model proposes a purely quantitative explanation in which the existence of a fixed critical structural threshold (i.e., identical for all individuals) is postulated and which qualifies the transition between normal and pathological aging (i.e., when the threshold is reached, inevitably the clinical signs manifest themselves). To respond to the more qualitative differences in the management of the individual’s resources, the CR model was proposed. The CR [52,53] corresponds to a more active functional model that refers to qualitative differences with respect to how the individual manages his or her resources. The CR model, unlike the cerebral reserve model, does not assume that there is a fixed threshold from which the functional disorder appears. This model focuses on the processes that allow individuals to sustain brain damage and maintain proper functioning. Two individuals with the same brain reserve can indeed distinguish themselves in the way they react to brain damage. In other words, if two individuals have the same amount of brain reserve, the individual with more CR can tolerate more extensive lesions before a clinical disorder appears than the other individual. The CR model is, in fact, used to interpret both brain alterations, due to brain damage, such as traumatic injury [54,55] or the normal aging process [51,52,53]^,^ and individual differences in the processing of information in the absence of brain injuries [47,48].

In its initial formulation, Stern distinguished two types of reserve: The CR and the compensation [52]. While cognitive reserve is limited to the individual differences observed in healthy subjects, compensation is associated with a specific response to brain damage. More precisely, Stern and collaborators referred to the CR to evoke an efficient use of strategies and functional connection networks between pre-existing neurons, similar to those used by healthy individuals [52]. Compensation instead indicated the use of alternative connection networks or new strategies not used by healthy individuals. The concept of CR has then undergone a significant evolution over the years, leading to the overcoming of this distinction. Recently Stern proposed the neural implementation of the CR and its two components: The neural reserve and neural compensation [53].

Neural reserve refers to individual differences, probably in the form of efficiency, capacity, or differential flexibility, in the use of networks of functional connections and cognitive processes in healthy subjects. An individual whose connection networks are more efficient or more flexible has a greater ability to use these networks and may be able to react if brain damage occurs.

Neural compensation refers, however, to individual differences in the use of functional connection networks and alternative cognitive processes as they are not used by healthy subjects. Neuronal compensation, understood as the ability to compensate connection networks and damaged processes, would, therefore, help to maintain or improve behavioral performance. On an empirical level, CR is currently operationalized through the use of brain visualization studies in which the similarities and differences in regional brain activation patterns and, above all, in connectivity patterns between brain regions associated with the realization of a task are examined. The peculiarity of the studies with brain visualization techniques conducted on the reserve, compared to those on cognition in aging, lies in the interest on the individual differences in the connectivity patterns between the different regions and how these differences are associated with indicators of CR. In other words, the hypothesis would be that the CR is related to a generic functional connection network that is not specific to the task performed and that it can be solicited in various tasks. For example, in 2005, Stern and collaborators examined, in a PET study, the relationships between cognitive reserve and brain activation level during a recognition task in a low memory load and a high condition memory load [56,57]. In this study, the CR was quantified with a global score that considered the years of schooling and with IQ. The results showed that functionally connected regions changed their activation level according to the memory load condition and the age of the participants. In addition, young adults, with greater CR, showed a high activation of the regions directly involved in the neuro-functional network responsible for the increase in the memory load between the two conditions and a decrease in the other brain areas. This differential activation has been interpreted as a neuronal manifestation of the reserve. On the contrary, older adults with a greater reserve showed a decrease in activations in the areas involved due to the increase in the memory load and an increase in the other regions not involved. The authors, therefore, assumed an inefficient use of the neuro-functional network usually used by young adults, which, in the elderly, produces changes in the activation of the areas involved. The different level of activation in the elderly would, therefore, represent a neural compensation of the network used by young adults to maintain a certain level of performance in response to the neurophysiological losses observed with advancing in age [58]. Like the brain reserve, CR indicates the brain’s resilience to pathological brain damage. CR, in particular, focuses on how the brain uses damaged resources. This type of reserve thus represents an efficiency model, as opposed to the threshold model-cerebral reserve, thanks to which the task is processed using fewer resources and in order to produce fewer errors.

Although cerebral reserve and CR have been examined as two independent models, various evidences indicate that the two models are interdependent and connected to each other. Both types of reserve share the implicit postulate according to which the resources accumulated during life allow the individual to maintain adequate behavior from a functional point of view. These resources are the result of the brain’s ability to change, develop, and adapt, structurally and functionally, thanks to a dynamic and continuous interaction between biological and environmental influences.

Among the factors that favor reserve accumulation as a factor of resilience to biological processes related to aging, postponing the clinical manifestations of pathologies, such as dementia, there is high schooling [58] or a high IQ [59]. It can, therefore, be assumed that the educational factors and IQ, which protect from the onset of dementia, are also those that allow individuals to manage age-dependent changes more effectively. Although schooling has a protective role, it can also “hide” the onset of the pathology, which, when it occurs, has a much faster course than the decline in individuals with low schooling. In 2008, Bruandet and colleagues, for example, studying 670 AD patients and followed for more than three years, found that patients with a high or medium level of education had a more important and rapid cognitive decline than individuals with a low level of education [60]. The mechanisms by which the level of education acts as a protector (and low schooling as an unfavorable factor), therefore, require future studies to clarify their role in establishing the reserve.

Indeed, how to quantify the concept of CR in patients with dementia is a problem in clinical practice [61]. It is simplistic to use education as the only parameter. Indeed, several studies suggest that the concept of CR is the sum of schooling, occupational activity, general intelligence, leisure activity, socialization, and physical activities [62]. 

There is no validated test or questionnaire for AD patients to quantify CR in all these aspects, above all for difficulty of administration, especially in the most advanced stages of the disease. Indeed, the caregiver does not always know how to provide detailed information referring to the past of the patient.

Several studies hypothesize pathophysiological mechanism underlying the concept of CR, focusing attention on three concepts: Neuroplasticity, neurogenesis, and locus coeruleus/noradrenergic system [62].

The concept of neuroplasticity refers to modifications in the neuronal and synaptic pathways in the central nervous system. In particular, the hippocampus and entorhinal cortex seem to have a period of greatest increase in AD patients with greater cognitive reserve [63].

The neurogenesis is a concept born in 1962 and defined as the ability to produce new neurons, especially in hippocampal dentate gyrus in adults. The damage of the neurogenesis can be highlighted in the early stages of Alzheimer disease and, therefore, could be studied as new disease markers [64,65].

Finally, several studies have attempted to study the relationship between AD and the locus coeruleus/noradrenergic system. A positive association between neuronal loss in locus coeruleus and the age at onset of the disease seems to be. The noradrenergic system also appears to have a neuroprotective effect, thanks to anti-inflammatory mechanism [66,67].

However, in several recent works the concept of CR has been better investigated [68,69], with the creation of new terms, in particular, the concept of resistance versus resilience^71^. Resistance would represent the ability to oppose AD pathology, while resilience would represent the ability to compensate for pathological damage [70]. In particular, resistance to AD would imply absence or lower than expected AD pathology, while resilience to AD would imply higher than expected AD pathology. Lifestyle factors, vascular risk, sleep, sex, and genetics could contribute to resistance and resilience. Mechanisms specific to resistance are passive (brain reserve and maintenance) and active (glucose metabolism and functional networks), and those mechanisms could interact with Aβ and tau clearance. Mechanisms specific to resilience are active (glucose metabolism and functional networks) [70]. 

In addition to the concepts of resistance and resilience, the concept of compensation was born. This term indicates neural recruitment that enhances cognitive performance. It is necessary to distinguish the different mechanisms of compensation: Upregulation, selection, and recruitment of additional processes [71]. Compensation by upregulation implies a greater neuronal recruitment in adults than in young people. Compensation by selection implies a neuronal recruitment as effective as possible (at less energy expenditure) in young people more than in adults. In the end, compensation by reorganization occurs when adults use an alternative neural mechanism, that is not available to younger individuals, to respond to aging-induced losses [71]. 

In our previous study [72], we examined the interaction between education (a proxy of CR) and specific measures of brain pathology (CSF Aβ_1-42_ and brain metabolism). 

Twenty-seven patients with probable AD, according to the National Institute of Neurological and Communicative Disorders and Stroke and by AD and Related Disorders Association diagnostic criteria, were consecutively recruited from the Clinic of Nervous System Diseases of OORR Foggia (Italy) between September 2012 and June 2013 (see Table 2).

We divided them according to years of formal education into two groups: High Education AD (HE-AD) patients (if median ≥ 5) versus Less Education AD (LE-AD) patients (if median < 5). HE-AD and LE-AD patients were similar for demographic and clinical characteristics and years of disease onset. 

In all patients, alternative neurological and psychiatric diagnoses were excluded, based on blood tests (complete blood count; liver, kidney, and thyroid function tests; serum cobalamin and folate; syphilis serology) and conventional MRI (T2- and T1-weigthed images, fluid-attenuated inversion recovery (FLAIR) images, diffusion-weighted images). After recruitment, patients underwent a detailed neuropsychological evaluation, lumbar puncture, and brain 18FDG-PET.

CSF was aliquoted in polypropylene tubes and temporarily stored at −22 °C to quantify, within one month after collection, Aβ_1-42_, 181p-tau, and Tau concentrations (Innotest ELISA; Innogenetics, Ghent, Belgium). PET images were reconstructed in repetitive Recon or Osem mode, processed with a statistical software 3DSSP, and evaluated by two independent nuclear specialists.

Differences in regional glucose uptake were investigated in the two groups using a two-sample *t* test design in SPM8, in order to evaluate the interaction between education levels and brain metabolism. Statistical tests were all performed considering the whole brain, and results were considered as statistically significant for *p* values < 0.05. 

The interaction between CR and CSF biomarkers on FDG-PET metabolism showed significant results when considering β-amyloid levels only. Selective modulation of CR on brain metabolism was found in the inferior and medial temporal gyrus bilaterally (Figure 1). Such interaction was driven by a negative association between CSF β-amyloid levels (inversely correlated to amyloid plaque deposition) and glucose uptake in HE-AD patients, and a direct association in LE-AD patients. The use of parameters, such as CSF Aβ_1-42_ and PET glucose metabolism, allowed us to consider both the active and the passive component of the CR concept. The main finding of the current study was the interaction between patients’ CR and β-amyloid deposition on metabolism in the medial temporal lobes, which have been previously shown as particularly sensitive to Aβ_1-42_ level [73,74]. In HE-AD patients, CR was inversely associated with glucose uptake, whereas the opposite effect was observed in LE-AD patients. This is in line with the CR theory.

## 4. Conclusions

Our findings suggest that, at least in early/moderate stages of AD, CR compensates neurodegeneration and allows the maintenance of patients’ cognitive performance, as previously reported by other authors [75,76,77,78]. Therefore, the full elucidation of the concept of CR could lead to the promotion of interventions aimed at slowing cognitive aging or reducing the risk of dementia. For this reason, it is necessary to create a standard measure of CR.

## Figures and Tables

**Figure 1 ijms-21-06279-f001:**
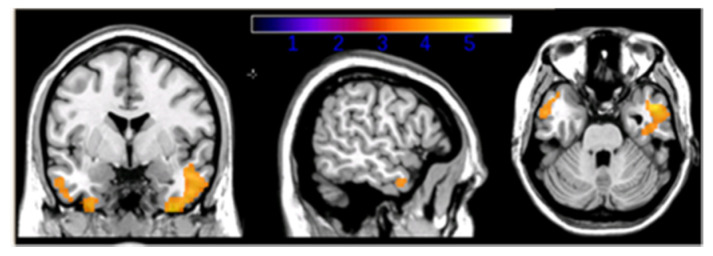
Interaction between education and Aβ1-42 values in AD patients’ groups, *p* < 0.05 level [72].

**Table 1 ijms-21-06279-t001:** Biomarkers and their changes in Alzheimer disease (AD).

Biomarkers	Changes in AD	
Aβ_1-42_	Marked reduction in AD	Reduction CSF Aβ_1-42_ is gold standard for AD; low CSF Aβ1-42 is found in Lewy bodies dementia
181p-tau	Marked increase in AD	High CSF 181P-tau is not specific for AD
T-tau	Marked increase in AD	High CSF T-tau is found in stroke, trauma and encephalities; very high CSF T-tau is found in Creutzfeld-Jakob

**Table 2 ijms-21-06279-t002:** Clinical and biochemical features of studied population level [72].

	HE-AD (n = 12)	LE-AD (n = 15)	*p* Value
Mean (SD) age (years)	68.6 (7.2)	73.9 (7.6)	n.s.
Sex (female|male)	8|4	10|5	n.s.
Mean (SD) MMSE score	17.5 (6.2)	17.9 (4.7)	n.s.
Mean (SD) ADL score	4.3 (0.9)	3.8 (1.3)	n.s.
Mean (SD) IADL score	3.4 (1.8)	3.8 (1.9)	n.s.
Mean (SD) GDS score	5.25 (3.4)	7.5 (4.5)	n.s.
Mean (SD) years of formal education	10.4 (2.5)	4.7 (0.7)	*p* < 0.001
Mean (SD) of Aβ_1-42_ values (pg/ml)	465.8 (140)	534.8 (183.6)	n.s.
Mean (SD) of 181P-tau values (pg/ml)	93.3 (65.6)	62.7 (39.8)	n.s.
Mean (SD) of T-tau values (ng/ml)	694.9 (570)	616.3 (461.7)	n.s.

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
