# Peer review of "Interaction between Cognitive Reserve and Biomarkers in Alzheimer Disease"

_ijms, 2020, doi:10.3390/ijms21176279_

Round 1

Reviewer 1 Report

Authors present a good review of various interactions between cognitive mechanisms and biomarkers in AD. The manuscript is well-written to the most of the part with a good flow. I have few topics that can be included in this review

a. comment on the new biomarker strategies in AD including exosomes, PET imaging of microtubule and SV2A

b. what are the potential and possible demerits of ATN scoring in early and late-stages AD diagnosis and therapeutic implications?

c. changes with other interventions with respect to metabolism and diet?

c. what could be the new research tools needed to completely understand and evaluate the interaction stages of cognitive functions in AD

d.Please correct the tenses (English grammar)- some of lines are in past and some are in past-perfect tense.

Author Response

a. Recent studies reported increseased P-tau concentrations in blood-borne neuron-derived exosomes. However, CNS tau efflux via exosomes is likely increased in Parkinson's disease but not in Alzheimer disease. 

The synaptic vescicle glycoprotein 2A (SV2A) combined with PET was used to quantify sinaptic density in the living human brain. The probe was sensitive enough to detect synaptic loss in patients with temporal lobe epilepsy. Its utility in AD remains to be established. 

b. ATN scoring is not usefull to establish early and late stages AD diagnosis and therapy.

c. Several studies explain the role of metabolism and diet in AD phisiopathology; but real role in AD and CR in AD remains to establish

d. A classification not only of disease but also of stadiation of gravity, and therapeutic strategy would be necessary 

e. I'll change tenses shortly and to re-upload to the file with these changes. 

Reviewer 2 Report

The authors studied the interaction between cognitive reserve and CSF biomarker levels in AD. The topic is very interesting.

There are some major and some minor issues raised:

Major

Line 287. Usually, CSF aliquots are stored at –80°C until testing, but they can be stored at –20°C if tested within 2 months. Please, explain if this is a typo error and the temperature is –22°C. Keeping them at +22°C for a few weeks may significantly affect biomarker levels (especially Aβ42).

What was the biomarker profile of the patients? Where all biomarkers abnormal in all patients (A+T+N+). Did some patients have low Aβ42 with normal phospho-tau? Please describe.

Minor

Line 38-39. Phospho tau is a marker of neurofibrillary tangle formation, but total tau is not. It is a nonspecific marker of neuronal/axonal destruction, including neurodegeneration (not necessarily tangle-related). Please change appropriately (the authors describe this correctly in the next paragraph concerning the ATN system)

Line 277. Please state the criteria according to which the diagnosis of probable AD was made.

Author Response

I will report the requested changes in a new file, that I'll upload shortly

Reviewer 3 Report

Authors discussed on the interactions between Cognitive Reserve and biomarkers in AD, which would be beneficial for many scientists. However, the compilation of biomarkers is not concrete and unorganized. Authors should make table of all biomarker for their claims and correlations. Authors need to standardize the definition and criteria of CR.

Author Response

I'll add the required table

Round 2

Reviewer 2 Report

At AUTHOR’S RESPONCE it is stated that “I will report the requested changes in a new file, that I'll upload shortly”. However, I could not find (or, at least I was not able to download) the authors’ answers. I do believe that a point-to-point answer to reviewer’s questions/comments is always helpful. Anyway, the revised manuscript is substantially improved, so I suggest to accept in the current form.

Author Response

Thank you

Reviewer 3 Report

As mentioned in the previous review, authors did not include many of the latest references and revisions as suggested.

Author Response

I have proceeded to add other recent references
